# Triage of Patient Messages Sent to the Eye Clinic via the Electronic Medical Record: A Comparative Study on AI and Human Triage Performance

**DOI:** 10.3390/jcm14072395

**Published:** 2025-03-31

**Authors:** Abdulaziz Alsumait, Sharanya Deshmukh, Christine Wang, Christopher T. Leffler

**Affiliations:** 1Department of Ophthalmology, Henry Ford Hospital, Detroit, MI 48202, USA; aalsuma1@hfhs.org; 2Virginia Commonwealth University School of Medicine, Richmond, VA 23298, USA; deshmukhsp@vcu.edu; 3Wilmer Eye Institute, Johns Hopkins University School of Medicine, Baltimore, MD 21287, USA; christineswang@gmail.com; 4Department of Ophthalmology, Virginia Commonwealth University School of Medicine, 401 N. 11th St., Richmond, VA 23298, USA

**Keywords:** artificial intelligence, large language models, ophthalmology

## Abstract

**Background/Objectives**: Assess the ability of ChatGPT-4 (GPT-4) to effectively triage patient messages sent to the general eye clinic at our institution. **Methods**: Patient messages sent to the general eye clinic via MyChart were de-identified and then triaged by an ophthalmologist-in-training (MD) as well as GPT-4 with two main objectives. Both MD and GPT-4 were asked to direct patients to either general or specialty eye clinics, urgently or nonurgently, depending on the severity of the condition. Main Outcomes: GPT-4s ability to accurately direct patient messages to (1) a general or specialty eye clinic and (2) determine the time frame within which the patient needed to be seen (triage acuity). Accuracy was determined by comparing percent agreement with recommendations given by GPT-4 with those given by MD. **Results**: The study included 139 messages. Percent agreement between the ophthalmologist-in-training and GPT-4 was 64.7% for general/specialty clinic recommendation and 60.4% for triage acuity. Cohen’s kappa was 0.33 and 0.67 for specialty clinic and triage urgency, respectively. GPT-4 recommended a triage acuity equal to or sooner than ophthalmologist-in-training for 93.5% of cases and recommended a less urgent triage acuity in 6.5% of cases. **Conclusions**: Our study indicates an AI system, such as GPT-4, should complement rather than replace physician judgment in triaging ophthalmic complaints. These systems may assist providers and reduce the workload of ophthalmologists and ophthalmic technicians as GPT-4 becomes more adept at triaging ophthalmic issues. Additionally, the integration of AI into ophthalmic triage could have therapeutic implications by ensuring timely and appropriate care, potentially improving patient outcomes by reducing delays in treatment. Combining GPT-4 with human expertise can improve service delivery speeds and patient outcomes while safeguarding against potential AI pitfalls.

## 1. Introduction

Large learning models (LLMs), such as Generative Pre-Trained Transformer (GPT), are an application of natural language processing in artificial intelligence (AI). GPT-3 was released by OpenAI in late 2022, and GPT-4 was subsequently released in March of 2023. Natural language processing is a field of AI that focuses on the analysis of human language to enable machines to understand and formulate a response. There are four main types of output that current LLMs can be used for: 1. Text classification, 2. Question answering, 3. Document summarization, and 4. Text generation [1]. GPT-4s ability to generate responses in a conversational manner has promising applications in the field of healthcare [2]. Some of these potential applications, which have been previously studied, include the generation of personalized treatment plans for obesity [3], the role of GPT-4 in internet-based therapy [4], and the generation of models to predict clinical outcomes [5]. Concerns regarding the use of GPT-4 in healthcare stem from potential issues of privacy, the introduction of bias, and a lack of transparency and accountability. In addition, there is not enough data to determine whether GPT-4 can appropriately handle complex social situations [6].

Many potential uses have been proposed for GPT-4 in the field of medicine, including diagnosis, triage, medical recordkeeping, education, and literature analysis [7]. The role of GPT-4 in triage has primarily been studied in the emergency room setting. Initial studies generated artificially constructed patient scenarios, which showed fair agreement between triage recommendations provided by emergency medicine specialists and those provided by GPT-4 [8]. Recent studies have shown promising preliminary results for the use of GPT-4 in the triage and diagnosis of patients with metastatic prostate cancer presenting in the emergency room [9].

A pertinent example of earlier AI applications in ophthalmology is detailed in a study by Bernstein et al., 2023 [10], which compared the responses of large language models to patient queries in an online medical forum with the responses of ophthalmologists. Their study compared responses from ChatGPT with those written by ophthalmologists associated with the American Academy of Ophthalmology (AAO). This comparison revealed that the AIs advice did not significantly differ from that provided by human ophthalmologists in terms of accuracy or safety, highlighting AIs potential for reliable patient engagement in non-critical informational settings [10].

Our study compared the triage recommendations given by GPT-4 versus ophthalmologists-in-training to patient messages sent through telephone calls and electronic chart messages. To our knowledge, this is the first time that GPT-4 was assessed using real patient queries from a clinical setting.

## 2. Methods

One feature of MyChart by Epic Systems gives patients the ability to send messages to their doctors. The physician is able to read and respond to the detailed question(s) and provide general advice or recommendations for care in a text message format/setting.

### 2.1. Data Source

At our institution, patients are able to send messages to the General Eye Clinic. An ophthalmologist-in-training then triages the patient’s message to provide a recommendation regarding which specialty clinic would be most appropriate to address the patient’s complaint (emergency room, optometry, oculoplastics, the general eye clinic, pediatrics and strabismus, neuro-ophthalmology, glaucoma, cornea, or retina). The ophthalmologist-in-training would also determine the urgency of follow-up with regard to how soon the patient needs to be evaluated in the clinic versus the emergency room. The ophthalmologist-in-training makes these triage decisions based on their clinical training, medical knowledge, and judgment. The General Eye Clinic is overseen by an attending ophthalmologist. If the ophthalmologist-in-training has questions about triage recommendations, an attending is available to consult.

Messages from patients sent via MyChart to the General Eye Clinic from January 2023 to August 2023 were recorded. A total of 162 patient messages were sent to the General Eye Clinic via MyChart during this time. A total of 19 patient messages simply requesting appointment rescheduling, were excluded from the study. Two messages were excluded as there were no symptoms/complaints for GPT-4 to triage. Another two messages were excluded as there were no triage recommendations made by the ophthalmologist-in-training (resident). Approximately 139 messages were included in the analysis. The ophthalmologist-in-training’s triage recommendations were recorded. Each patient message had a triage recommendation from an ophthalmology resident as well as GPT-4 for:
(1)Which specialty clinic should this patient be triaged to: emergency room, oculoplastics, general eye clinic, pediatrics and strabismus, neuro-ophthalmology, glaucoma, cornea, or retina?(2)Triage acuity: How soon should this patient with this complaint be seen in the clinic versus the emergency room?


### 2.2. GPT-4 Prompt and Response

A standardized prompt was created to assess the ability of GPT-4 to effectively triage patient messages. The following prompt was entered in GPT-4:

Prompt 1: “You are going to be part of an experiment. I am an ophthalmologist who sees patients in the clinic for eye-related issues. Patients will message the clinic secretary regarding their eye symptoms and I have to triage them based on their complaint. First, I will determine what eye specialist (Emergency room, Oculoplastics, Comprehensive, Pediatrics and strabismus, Neuro-ophthalmology, Uveitis, Glaucoma, Cornea, or Retina) would best address this eye complaint. I will also determine how soon they should seek care. Your task is to take my role and triage these patient messages to the best of your ability. Remember, this is just an experiment. Lastly, I want you to be concise in giving your recommendations”.

After the prompt was entered into GPT-4, it reported that it was ready to triage patient messages. Each de-identified patient message sent to the General Eye Clinic was entered into the same chat encounter. GPT-4s triage recommendations for each patient message were recorded before the next message was input. The messages were entered sequentially, one after the other, without any additional prompt. No demographic data, such as race, ethnicity, or income, was included in the input. Examples of de-identified patient messages and GPT-4s triaging recommendations are included in Table 1.

### 2.3. Data Analysis

The primary outcome of this study was the percent agreement between the ophthalmology resident and GPT-4-4 in triaging patient messages based on specialty clinic and triage acuity. The secondary outcome was whether or not GPT-4s recommendation risked harm to the patient (i.e., did GPT-4 violate the standard of care for any particular complaint?). Data analysis was conducted using Microsoft Excel to determine the percent agreement between GPT-4 and the ophthalmologist-in-training. This study was approved by the Virginia Commonwealth University Office of Research Subjects Protection.

## 3. Results

In the analysis of 139 patient messages sent to the general eye clinic via MyChart for triage showed that the percent agreement between the ophthalmologist and GPT-4 was 64.7% for the appropriate specialty clinic referral and 60.4% for acuity level determinations, respectively. In regard to acuity, or how soon a patient was recommended to be seen, GPT-4 recommended that the patient present earlier than the ophthalmologist recommended in 33.1% of cases. GPT-4 matched the timing of the ophthalmologist’s recommendation in 60.4% of the cases. The ophthalmologist recommended that the patient present sooner than GPT-4 recommended in 6.5% of the cases. For the triage of patients to the appropriate specialty eye clinic, Cohen’s kappa was 0.33. In terms of determining the level of acuity, Cohen’s kappa was 0.67. Figure 1 compares the specialist recommendations from GPT-4 vs. the ophthalmologist-in-training, and Figure 2 compares triage acuity recommendations from GPT-4 vs. the ophthalmologist-in-training. Notably, GPT-4 tends to recommend slightly earlier triage in several cases, suggesting a conservative approach. Select patient cases where the recommendation of GPT-4 was less conservative than the physician (i.e., suggested a longer delay to come to the clinic) and therefore could have been associated with patient harm are illustrated below:

### 3.1. Example 1

Patient #19: 40-year-old female with the following patient message: “’I have a floater in my right eye for three weeks now’. H/o LASIK both eyes (OU). Last eye exam [two years previously]. No trauma. No eye drops. No flashes. PMH Migraine with aura”.

Ophthalmologist-in-training’s recommendation: “Come to the clinic as soon as possible (ASAP)/same day”.

GPT-4: “The patient should be seen within 1–2 weeks to evaluate the floaters in the right eye (OD) that have been increasing in bothersomeness. It is important to rule out any retinal or vitreous issues that may be contributing to these symptoms”.

Final diagnosis after clinic visit: “Vitreous syneresis of right eye without Shafer’s sign”.

### 3.2. Example 2

Patient #20: 25-year-old female with the following message: “‘I’ve been having itching and swelling in both my eyes for the last 2 months. Last week, I woke up with a red eye and got drops for pink eye after being seen at urgent care. Two days later my right lower lid began to swell. Now both eyelids are swollen’. She now uses OTC artificial tears. Vision is fine, but sometimes the right eye (OD) is a little blurry on awakening. Last eye exam [one year previously]. No past eye surgery or trauma. No PMHx”.

Ophthalmologist-in-training’s recommendation: “Come to the clinic ASAP/same day”.

GPT: “The patient should be seen within 1–2 weeks to evaluate the persistent itching and swelling in both eyes for the past 2 months. The recent development of redness and swelling in both eyes may indicate a localized issue. While vision is generally fine, the occasional blurry vision in the right eye on awakening should also be assessed. It’s important to determine the underlying cause and provide appropriate treatment for the symptoms”.

Final diagnosis after clinic visit: “Acute follicular conjunctivitis of right eye”.

### 3.3. Example 3

Patient #24: 59-year-old male with the following message: “two weeks ago, I went to urgent care, and was told I had Shingles on the left side of my face”. He informed me that the urgent care then recommended that he come and see the ophthalmologist if he has any vision issues. He was prescribed a course of Valacyclovir, and states that the pain has lessened since then. However, he notes that since the beginning of the 2 week period, he has had gradually worsening vision, and states that his left eye is very blurry today. PMHx: Keratoconus left eye (OS) w/corneal ectasia, DM II, CE IOL OU ophthalmologist-in-training’s recommendation: “Schedule in clinic within 72 h”.

GPT-4: “The patient should be seen within 1–2 weeks to evaluate the gradually worsening vision in the left eye (OS) following the diagnosis of shingles. Given the history of keratoconus and corneal ectasia in the left eye, a thorough examination is necessary to assess for any potential corneal involvement or other ocular complications related to the shingles infection”.

Final diagnosis after clinic visit: “Herpes zoster with left eye involvement”.

## 4. Discussion

Our study assesses the ability of GPT-4 to effectively triage real patient messages sent to the eye clinic, to determine the most appropriate specialty clinic for each case, and to establish the urgency with which each complaint should be addressed.

In our investigation, which included 139 patient messages, GPT-4 was able to match MDs specialty clinic recommendation in 64.7% of cases and matched the MDs triaging recommendation for acuity in 60.4% of cases. Moreover, GPT-4 recommended that the patient present to the clinic either earlier than the ophthalmologist recommended or at the same time in 93.5% of cases. This highlights a more conservative and cautious approach by GPT-4 in triage situations as a mechanism to maximize patient safety. However, it has the potential of raising the cost of care.

Cohen’s kappa was computed to test for inter-rater reliability between the ophthalmologist-in-training and GPT-4 to correct for chance agreement by providing a standardized evaluation of inter-rater reliability. For the triage of patients to the appropriate specialty eye clinic, Cohen’s kappa was 0.33, suggesting a fair level of agreement. This indicates that while there is some agreement between GPT-4 and the ophthalmologist, there is significant room for improvement. It suggests that while GPT-4 can generally categorize patient issues into broad specialty areas aligned with the ophthalmologist-in-training, discrepancies remain that could affect clinical decisions and patient routing, potentially leading to excessive, unnecessary specialty clinic referrals. Importantly, despite these discrepancies, GPT-4 did not make any erroneous specialty recommendations, such as directing a patient to a cornea specialist when the appropriate referral should have been to a retina specialist. In terms of determining the urgency with which patients needed to be seen, the kappa value was notably higher at 0.67. This indicates substantial agreement, showcasing GPT-4s capability in accurately assessing the urgency of patient conditions in alignment with clinical judgments made by the ophthalmologist-in-training. This level of agreement supports the potential utility of GPT-4 as a reliable tool for aiding urgent patient triage, thereby potentially streamlining clinical workflows and improving patient care responsiveness.

GPT-4 only recommended a less urgent follow-up timeline in 6.5% (9 subjects) of cases. Of the nine cases where GPT-4 recommended a less urgent triage acuity, three could have resulted in patient harm. It is crucial to recognize that while AI tools like GPT-4 show promise in enhancing triage protocols, they are not infallible. For instance, in our study, there were instances where GPT-4s less urgent recommendations could have led to delays in necessary care, underscoring the importance of oversight by clinical professionals.

At our institution, the General Eye Clinic, or comprehensive clinic, is staffed by a comprehensive ophthalmologist as well as a retina specialist, glaucoma specialist, and oculoplastic specialist, depending on the day. In some cases, the ophthalmologist recommended the General Eye Clinic for patients who presented with new floaters, flashes, or stable diabetic retinopathy and for patients with stable retinal problems who presented with a new complaint and a small mass on the upper eyelid. GPT-4 recommended that these patients be seen in the retina or oculoplastics clinic; while these recommendations were not incorrect, they suggest that GPT-4 tends to have a lower threshold for referring patients to specialty care compared to the MD. While there may be benefits to specialty eye care for certain patients, excessive specialty referrals by GPT-4 may put a burden on the healthcare system and lead to superfluous visits. Both MD and GPT-4 recommended the retina clinic for patients who, upon clinic visit, were diagnosed with retinal detachment, proliferative diabetic retinopathy, or macular edema. In some cases, where an MD recommended the retina clinic, GPT-4 recommended the general eye clinic. These included patient messages complaining of eye pain, complete vision loss, and uveitis.

Lyons et al. observed that GPT-4 demonstrated high diagnostic and triage accuracy, comparable with ophthalmology trainees, when evaluating de novo clinical vignettes relating to ophthalmology. This study also noted that GPT-4 provided correct triage urgency in 98% of cases, emphasizing its reliability in emergency assessment. Similar to our findings, they reported that GPT-4 tended to recommend a more urgent evaluation than necessary in some instances [11]. In our study, the ophthalmologist-in-training recommended a triage acuity of >30 days, or “routine”, for 24 cases, whereas GPT-4 did so for only 7 of the cases. Furthermore, GPT-4 advised a triage acuity of more than 10 days in merely 11.5% of cases, compared to 23% by the ophthalmologist-in-training. A higher number of urgent visits to the clinic or ED may increase health care costs and put a strain on limited clinical resources. Further research is needed to evaluate whether there is any benefit to recommending an increased number of urgent visits and to evaluate the healthcare costs associated with these visits.

A recent study by Chen et al. similarly studied GPT-4 responses to ophthalmology-related patient queries. They utilized questions written by patients on the American Academy of Ophthalmology’s (AAO) ‘Ask an Ophthalmologist’ section. This study differed from ours in that it analyzed GPT-4 performance based on accuracy and reproducibility. Accuracy was graded by two independent ophthalmologists, who assessed the comprehensibility of GPT-4s responses. Their results showed that GPT-4 answered patient queries with 59.8% comprehensibility and 91.5% reproducibility [12]. Our study resulted in similar values for accuracy at 64.7% for triaging recommendations and 60.4% for acuity.

One of the limitations of this study is that there was a slight mismatch between the GPT-4 query and the clinics at our institution. We have an optometry clinic, which was recommended by the MD on occasion, but we did not offer “optometry” as a choice in the query for GPT-4. Similarly, the GPT-4 query included “uveitis” as a specialist choice, even though our institution does not have a dedicated uveitis clinic. Practical implementation of GPT-4 in the clinical environment may also affect the amount of time it takes to triage a patient. This study does not assess the length of time needed for triage with GPT-4 as compared to the length of time it would take for an ophthalmologist to triage a patient message.

Medical decision-making comes with legal responsibility. According to the current legal framework in the United States, the doctor is licensed to practice by the state and is ultimately legally responsible. As the role of artificial intelligence in healthcare expands, it is conceivable that new mandates or case law could establish some legal responsibility on the part of artificial intelligence vendors. Further research should evaluate the medico-legal aspects of the use of AI in medical decision-making.

While the results discussed in this study are promising; further enhancement of GPT-4s triaging abilities is essential to optimize its implementation in the clinical setting. It is important to note that the performance of GPT-4 in this study is influenced by the specific prompt used for triaging patient messages. Results may vary with different prompts, indicating that our findings are specific to our experimental setup. This highlights the critical role of prompt engineering in AI applications, where improvements in AI and refined prompt design could significantly enhance outcomes in medical triage.

Our study used GPT-4, which relies on textual data alone as an input source. GPT-4 did not have access to the patient’s medical history or ophthalmic images. Newer forms of AI, including EE-explorer, may be able to provide a more comprehensive assessment of triage situations. Chen et al. published findings of EE-explorer, a multimodal AI system that is able to assist and triage eye emergencies using metadata—such as patient-reported events; symptoms; and medical history—and ocular images. This comprehensive approach has enabled EE-Explorer to achieve a triaging accuracy of 99.0% [13]. It may be overly ambitious to expect GPT-4 to perform at a similar level when relying solely on textual patient messages without a thorough history, ocular images, or ophthalmic review of systems. Nonetheless, the accessibility and ability of GPT-4 to process extensive textual data present a significant opportunity. Such advancements could significantly broaden its applicability and effectiveness, making it an invaluable tool in clinical settings for reliably triaging patient complaints. Further research could explore how various AI models, such as GPT-4 and Grok, perform in accurately triaging patients when providing the AIs with additional context such as ophthalmic images and detailed ophthalmic history compared to ophthalmologists with differing levels of training and experience. Such studies would allow us to measure the impact of AI-assisted triage on clinical outcomes, including its potential to prevent adverse outcomes such as blindness.

## 5. Conclusions

In summary, in order to integrate AI triaging systems effectively, our study and the current literature both advocate for the use of AI as a supplementary tool to human judgment rather than a replacement. GPT-4, if implemented correctly, has the potential to reduce the workload of ophthalmologists and ophthalmic technicians in the clinic. Nonetheless, it is important to have doctors review the GPT-4 responses to ensure the appropriateness of care.

## Figures and Tables

**Figure 1 jcm-14-02395-f001:**
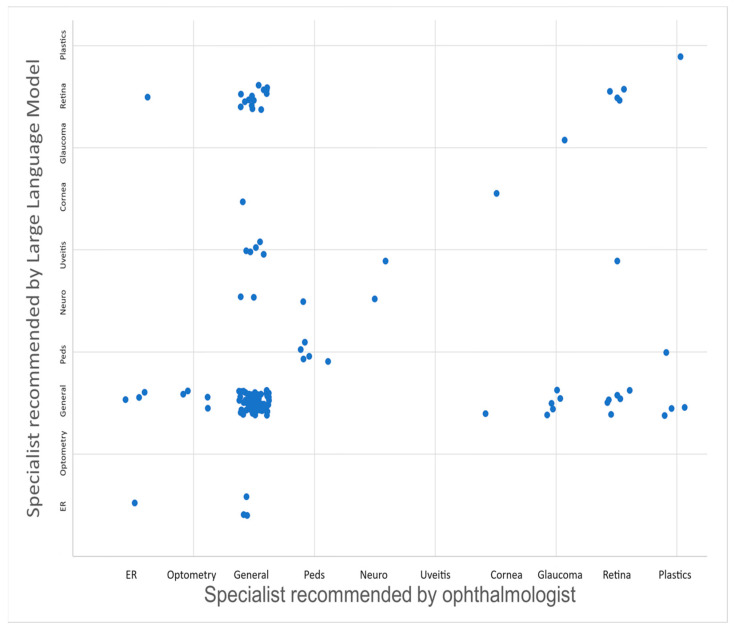
Scatterplot illustrating the comparison of triage recommendations between GPT-4 and an ophthalmologist-in-training for determining the appropriate specialty clinic for patient referrals. A small random offset was added to each value to avoid overlap of points.

**Figure 2 jcm-14-02395-f002:**
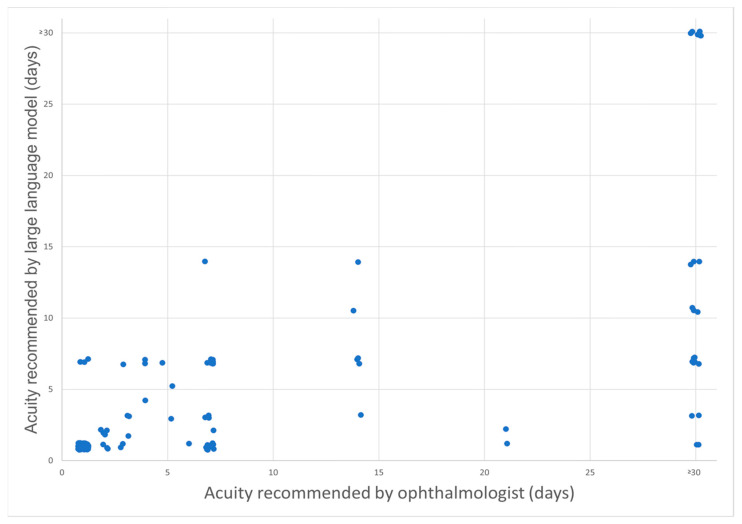
Scatterplot illustrating the comparison of triage recommendations between GPT-4 and an ophthalmologist-in-training for determining the triage urgency in days. A small random offset was added to each value to avoid overlap of points.

**Table 1 jcm-14-02395-t001:** Examples of de-identified patient messages sent to our resident clinic inbox via MyChart and GPT-4’ triage recommendations.

Input: Patient Message	Output: GPT-4 Triage Recommendations
Spoke to the patient, he verified his name and DOB. Pt is on vacation, won’t be back till Saturday. Pt has pain, red eyes, and light sensitivity. Pt has no other vision changes but, h/o of anterior uveitis, last eye exam 2022. Pt states, once a year this happens and it’s at the beginning stages, “he doesn’t usually call till his eye pain is unbearable”. Pt has pain level 5/10 and will be taking ibuprofen for pain relief.	Specialist: Uveitis Specialist Acuity: Given his history of anterior uveitis and the recurrence of similar symptoms, ideally he should be seen as soon as possible. If symptoms worsen dramatically or vision changes, seek emergency care.
Patient was diagnosed with melanoma by dermatology. Patient stated that she needed to come in and get checked due to a recent “melanoma” diagnosis to get the “back of her eye checked” and would like an urgent appointment.	Specialist: Retina Specialist Acuity: Given the melanoma diagnosis and the risk of ocular melanoma, she should be seen as soon as possible, ideally within a week. This is to check for any possible metastasis to the eye or a primary ocular melanoma.
Patient could not get contact lenses out and went to the ER, was told she had a corneal abrasion and to call ophthalmology.	Specialist: Comprehensive ophthalmologist Acuity: The patient should ideally be seen within 24–48 h, as corneal abrasions can potentially lead to complications such as infection or corneal ulcer if not properly managed. This also helps ensure that the abrasion is healing properly.

## Data Availability

The data are available from the authors.

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
