# Peer review of "Triage of Patient Messages Sent to the Eye Clinic via the Electronic Medical Record: A Comparative Study on AI and Human Triage Performance"

_jcm, 2025, doi:10.3390/jcm14072395_

Round 1
Reviewer 1 Report
Comments and Suggestions for Authors
The title “Comparative Evaluation of ChatGPT-4 and Ophthalmologist-2 in-Training in the Triage of Patient Messages Sent to the Eye 3 Clinic via the Electronic Medical Record” describes the study's main emphasis on comparing ChatGPT-4 with ophthalmology trainees in the triage of patient communications. Nonetheless, the title may be enhanced for clarity and precision. The authors should contemplate incorporating "A Comparative Study on AI and Human Triage Performance" to elucidate the research scope. The abstract is effectively organized by encapsulating the purpose, methods, results, and conclusion. Nonetheless, reference to the potential therapeutic implications of incorporating AI in ophthalmology triage would be advantageous. The conclusion must clearly articulate the integration of GPT-4 into healthcare operations to improve efficiency while safeguarding patient safety.
The introduction presents a compelling justification for assessing ChatGPT-4 in triage; nevertheless, it would be enhanced by a more comprehensive examination of current AI uses in ophthalmology. The authors should incorporate a concise paragraph that contrasts earlier AI-driven triage models with the positioning of ChatGPT-4 within this framework. Furthermore, incorporating previous research that investigated AI-driven triage in emergency contexts will enhance the background. Referencing a pertinent study like Bernstein et al., 2023, which juxtaposes AI responses with ophthalmologists' comments in online medical forums, would enhance context and underscore the study's importance.
The method section is thorough, outlining patient message collecting, GPT-4 prompting, and data analysis. Nevertheless, further details regarding the instructions provided to the ophthalmologists-in-training for triaging communications would improve clarity. It would be beneficial to clarify if inter-rater reliability testing was conducted among the ophthalmologists. Additionally, it is advisable to examine the constraints of GPT-4’s training data and their potential impact on triage accuracy. To enhance methodological rigor, the authors should reference Lyons et al., 2023, which assesses AI chatbot efficacy in ophthalmology triage.
The results are clearly articulated and illustrate GPT-4's performance in comparison to human triage. Further examination of instances when GPT-4's suggestions diverged from the ophthalmologists' conclusions is essential. The authors should clarify the therapeutic implications of GPT-4 suggesting less urgent triage in 6.5% of patients and whether this poses a safety risk. A brief paragraph addressing the potential ramifications of excessive triaging by GPT-4, which could result in superfluous specialty referrals, would be beneficial. Referencing Chen et al., 2024, which examines the accuracy of AI in ophthalmic instances, would enhance the discourse.
The discussion section adeptly situates the findings within the wider framework of AI triage literature. The authors must openly acknowledge the limits of GPT-4, especially its incapacity to evaluate non-verbal cues or supplementary patient information beyond textual input. Incorporating a paragraph on prospective research avenues, such as the integration of multimodal AI systems that amalgamate imaging data with text-based triage, would be beneficial. The authors should also examine how GPT-4 could be enhanced using reinforcement learning incorporating feedback from human ophthalmologists. Citing EE-Explorer by Chen et al., 2023, which uses multimodal AI for ophthalmic triage, would enhance this topic.
The conclusion accurately summarizes the results; nevertheless, it can place greater emphasis on the actual use of GPT-4 within clinical environments. The authors should contemplate incorporating a comment regarding the integration of AI models such as GPT-4 into electronic medical record (EMR) systems to support ophthalmologists. Furthermore, the necessity for ongoing supervision and modification of AI models should be underscored. Citing Gebrael et al., 2023, which examines AI-assisted triage in emergency departments, would underscore the wider ramifications of AI in medical triage.
The table and figures offer significant insights into the comparative efficacy of GPT-4 and human triage. Figures 1 and 2 would benefit from supplementary annotations elucidating substantial trends. A concise analysis within the primary text encapsulating essential insights from these figures would improve reader understanding.
The article provides a significant investigation of the role of AI in ophthalmic triage. Although the results are encouraging, it is essential to meticulously evaluate GPT-4's limits prior to endorsing its extensive clinical application. Furthermore, examining the potential evolution of GPT-4’s performance through ongoing upgrades and training would be advantageous.
Author Response
To the reviewer(s),
Thank you very much for your constructive feedback on our manuscript. We greatly appreciate the time and effort you dedicated to reviewing our work. Your insights have been invaluable in guiding us towards enhancing our manuscript on the Triage of Patient Messages Sent to the Eye Clinic via the Electronic Medical Record: A Comparative Study on AI and Human Triage Performance. We look forward to improving our manuscript for potential publication in your esteemed journal. All new changes in the manuscript can be found in red font.
Sincerely,
Alsumait A, Deshmukh S, Wang C, Leffler C
Comments
1. Title: may be enhanced for clarity and precision. The authors should contemplate incorporating "A Comparative Study on AI and Human Triage Performance" to elucidate the research scope.
We have enhanced the title to: Triage of Patient Messages Sent to the Eye Clinic via the Electronic Medical Record: A Comparative Study on AI and Human Triage Performance‘
2. Abstract: reference to the potential therapeutic implications of incorporating AI in ophthalmology triage would be advantageous.
We agree with the reviewer’s comment and have added this to the abstract (lines 33-38)
3. Conclusion must clearly articulate the integration of GPT-4 into healthcare operations to improve efficiency while safeguarding patient safety.
We agree with the reviewer’s comment and have added this to the abstract (lines 38-40)
4. Introduction: more comprehensive examination of current AI uses in ophthalmology. The authors should incorporate a concise paragraph that contrasts earlier AI-driven triage models with the positioning of ChatGPT-4 within this framework.Referencing a pertinent study like Bernstein et al., 2023, which juxtaposes AI responses with ophthalmologists' comments in online medical forums, would enhance context and underscore the study's importance.
We agree with the reviewer’s comment and have added this to the introduction (lines 65-72)
5. Methods: details regarding the instructions provided to the ophthalmologists-in-training for triaging communications would improve clarity. It would be beneficial to clarify if inter-rater reliability testing was conducted among the ophthalmologists.
At our institution, only one ophthalmologist-in-training is triaging patient messages that come to the clinic at a time. The ophthalmologist-in-training makes these triage decisions based on their clinical training, medical knowledge, and judgment. The clinic is overseen by an attending ophthalmologist. If the ophthalmologist-in-training has questions about triage recommendations, an attending is available to consult. These details have been added to the methods section under section 2.1 (lines 90-93).
6. Clarify the therapeutic implications of GPT-4 suggesting less urgent triage in 6.5% of patients and whether this poses a safety risk.
The instances where GPT-4 suggested a less urgent triage recommendation were presented in the results section (lines 153-201). The implications were also discussed in the discussion (para 4 lines 240-246)
7. Potential ramifications of excessive triaging by GPT-4, which could result in superfluous specialty referrals, would be beneficial. Referencing Chen et al., 2024, which examines the accuracy of AI in ophthalmic instances, would enhance the discourse.
We agree with the reviewer’s comment and have added this to the discussion (lines 229-232 AND 270-274)
8. Acknowledge the limits of GPT-4, especially its incapacity to evaluate non-verbal cues or supplementary patient information beyond textual input. Incorporating a paragraph on prospective research avenues, such as the integration of multimodal AI systems that amalgamate imaging data with text-based triage,
We agree with the reviewer’s comment and have added this to the discussion (lines 300 and on)
9. Conclusion: necessity for ongoing supervision and modification of AI models should be underscored. Citing Gebrael et al., 2023, which examines AI-assisted triage in emergency departments, would underscore the wider ramifications of AI in medical triage
Our conclusion concurs with the reviewer’s comment necessitating further supervision of AI models in healthcare, stating that it should be a supplementary tool to human judgment, underscoring the potential harmful pitfalls of AI currently.
10. Figures #1 and #2 would benefit from supplementary annotations elucidating substantial trends. A concise analysis within the primary text
-
- Figure #1: Supplementary annotation lines 249-255 describe some of the trends for figure #1
- Figure #2: Added supplementary annotation lines 152-153
Reviewer 2 Report
Comments and Suggestions for Authors
1. The study only checks agreement between GPT-4 and the ophthalmologist-in-training. It does not check if the triage decisions are correct. Just because the AI agrees with a human does not mean it is good at triage. The authors should check patient outcomes to see if GPT-4 makes the right triage decisions. If they cannot do this they should write in the paper that this is a limitation.
2. The study does not use strong statistical tests. It only gives percentage agreement but does not check if this agreement is by chance. A Cohen’s kappa statistic should be used to measure if the agreement is real. Without this, the results are weak. The authors should add this test.
3. The sample size is small with only 139 messages. This is not enough for strong conclusions. The results may change with more data. The authors should do a power analysis to check if the sample size is enough. If they do not they must say in the paper that the study may not have enough data to be sure of the results.
4. GPT4 is compared to only one ophthalmologist-in-training. This makes the study weak because one person is not enough to be a gold standard. Different doctors may triage differently. The authors should have used multiple experts and checked how much they agree with each other. If they do not fix this they must write that using only one doctor makes the study weak.
5. The study does not compare GPT4 with other AI systems. There are better AI models for triage. Some like multimodal AI that uses images and text together. GPT-4 only reads text. This could not enough for eye problems. The authors should compares GPT4 to models like EE-Explorer. If they do not do this they must say in the paper that they did not use the best AI technology or cite others for future research.
6. GPT-4 does not have access to past patient records and images. This is a problem because eye doctors use past records and images to make decisions. The authors should explain how this limits the AI’s ability. If they do not fix this they must write as limitation that the AI does not have enough data to make the best triage choices.
7. The study does not check for bias in GPT4’s decisions. AI can have bias and make mistakes for some groups of patients. The authors should analyze if GPT-4 makes different triage decisions based on patient demographics. If they do not they must say in the paper that they did not check for bias which is a big problem.
8. The study does not measure if GPT4’s triage recommendations help reduce doctor workload. AI is supposed to make triage easier but the paper does not check if it actually helps. The authors should measure if using GPT4 saves doctors time. If they do not they must say in the discussion that they did not check if the AI is useful in real practice.
9. The study does not check if the triage recommendations increase healthcare costs. If GPT4 sends too many patients to urgent care, it may increase costs instead of helping. The authors should measure if AI triage is cost-effective. If they do not they must say in the paper that they did not check if the AI makes healthcare more expensive as limitation of the study.
10. There is no discussion of legal risks. If GPT4 makes a wrong triage decision and a patient is harmed, who is responsible? The study does not talk about this. The authors should explain the legal problems of using AI in triage. If they do not they must write that the study does not check the legal risks of AI mistakes.
11. The authors should fix these problems or clearly say in the paper that the study has these limitations. If they do not the results should not be fully trusted and the AI should not be used for real patients.
Author Response
To the reviewer(s),
Thank you very much for your constructive feedback on our manuscript. We greatly appreciate the time and effort you dedicated to reviewing our work. Your insights have been invaluable in guiding us towards enhancing our manuscript on the Comparative Evaluation of ChatGPT-4 and Ophthalmologist-in-Training in the Triage of Patient Messages Sent to the Eye Clinic via the Electronic Medical Record. We look forward to improving our manuscript for potential publication in your esteemed journal. All new changes in the manuscript can be found in red font.
Sincerely,
Alsumait A, Deshmukh S, Wang C, Leffler C
Comments:
- The study only checks agreement between GPT-4 and the ophthalmologist-in-training. It does not check if the triage decisions are correct. Just because the AI agrees with a human does not mean it is good at triage. The authors should check patient outcomes to see if GPT-4 makes the right triage decisions. If they cannot do this they should write in the paper that this is a limitation.
Our analysis primarily focused on assessing the agreement between GPT-4 and an ophthalmologist-in-training in triaging patient messages, rather than directly evaluating the clinical outcomes of these triage decisions.
The primary objective of this study was to determine whether AI, specifically GPT-4, is capable of mimicking the triage capabilities of a human ophthalmologist to a degree that would justify further, more detailed investigations. As such, our current work serves as a preliminary step, establishing a baseline of AI performance dealing with real human patients, sending messages to the eye clinic to be triaged in a controlled setting. The use of real world scenarios, esspecially patient messages has not been explored in the ophthalmology literature to our knowledge.
We acknowledge the critical importance of evaluating the clinical efficacy of AI triage through patient outcomes. Ideally, future studies would involve randomized controlled trials that assign patient messages to different triage protocols involving various AI models and ophthalmologists with different levels of training. Such studies would allow us to measure the impact of AI-assisted triage on clinical outcomes, including its potential to prevent adverse outcomes such as blindness. However, this would require a substantially larger sample size and more complex study designs to achieve the necessary statistical power to detect clinically significant differences. See lines 312-318 for updates to the manuscript regarding this comment.
- The study does not use strong statistical tests. It only gives percentage agreement but does not check if this agreement is by chance. A Cohen’s kappa statistic should be used to measure if the agreement is real. Without this, the results are weak. The authors should add this test.
We agree with the reviewer and thank them for this suggestion. Cohen’s kappa was computed to test for inter-rater reliability between the ophthalmologist-in-training and GPT-4 as suggested by the reviewer. Please see lines 147-149 in the results section as well as lines 221-239 in the discussion.
- The sample size is small with only 139 messages. This is not enough for strong conclusions. The results may change with more data. The authors should do a power analysis to check if the sample size is enough. If they do not they must say in the paper that the study may not have enough data to be sure of the results.
A power analysis was conducted to check if the sample size is enough. Please see below:
- Expected Agreement Rate (P1): 70% or 0.70
- Null Hypothesis Agreement Rate (P0): 50% or 0.50
- Alpha (α): 0.05 (for a 95% confidence level)
- Power (1 - β): 80% or 0.80
- Test Type: Two-sided
The required sample size is approximately 90 patients to achieve 80% power in detecting a 20% difference in agreement rates between GPT-4 and the human ophthalmologist, with a significance level of 5% in a two-sided test
- GPT4 is compared to only one ophthalmologist-in-training. This makes the study weak because one person is not enough to be a gold standard. Different doctors may triage differently. The authors should have used multiple experts and checked how much they agree with each other. If they do not fix this they must write that using only one doctor makes the study weak.
Here GPT4 was not compared with one or more doctors responding in a simulated or test situation. GPT4's response was compared to the actual response delivered by the practitioners in the department of ophthalmology in real time. One limitation of our study is that it is not possible to have patients call multiple different university ophthalmology departments to look at the variation in responses between departments, because established patients typically only have a relationship with one ophthalmology practice at a time.
- The study does not compare GPT4 with other AI systems. There are better AI models for triage. Some like multimodal AI that uses images and text together. GPT-4 only reads text. This could not enough for eye problems. The authors should compares GPT4 to models like EE-Explorer. If they do not do this they must say in the paper that they did not use the best AI technology or cite others for future research.
GPT4 has the ability to analyse images if they are uploaded. We did not do that however and relied on textual data only. This is mentioned as a limitation in the last paragraph of the discussion (lines 300-313). In that same paragraph we also talk about other AI systems in triaging eye problems specifically EE-explorer (lines 303-308).
- GPT-4 does not have access to past patient records and images. This is a problem because eye doctors use past records and images to make decisions. The authors should explain how this limits the AI’s ability. If they do not fix this they must write as limitation that the AI does not have enough data to make the best triage choices.
We agree with the reviewer, as such this is mentioned as a limitation in our study (lines 300-303 and 308-310)
- The study does not check for bias in GPT4’s decisions. AI can have bias and make mistakes for some groups of patients. The authors should analyze if GPT-4 makes different triage decisions based on patient demographics. If they do not they must say in the paper that they did not check for bias which is a big problem.
Inputs into GPT-4 did not include patient demographic data such as race or ethnicity. There was no way for the AI to know who sent the message to the clinic. Example of an input into GPT-4:
“Patient says he is having eye pain in his left eye. Says the eye is still red, his vision is still blurry, and his left nostril keeps running. His eye also runs with tears. He is not sure if it's infected.Patient had PCIOL Sx one year ago then 2-3 days later eye became light sensitive, with tearing the left eye excessively. Denies pain, but says it is hard to open eye in the morning, VA has not cleared totally since having surgery as well. No flashes, floaters, or curtain OS. Pt using artificial tears daily OS to treat with some success.”
Clarification that no demographic data was included into the input to GPT-4 was added to the methods section (lines 125-126).
- The study does not measure if GPT4’s triage recommendations help reduce doctor workload. AI is supposed to make triage easier but the paper does not check if it actually helps. The authors should measure if using GPT4 saves doctors time. If they do not they must say in the discussion that they did not check if the AI is useful in real practice.
This project was intended to compare GPT-4’s triaging recommendation to that of the current ‘golden standard’ which is a licensed physician. We shed light on the potential benefits of AI in triaging ophthalmic complaints including potentially saving doctors time in the future once this AI system is more robust and able to be trusted to make such decisions. We did not measure if this saves doctors time currently. This was stated as a limitation (lines 288-292).
- The study does not check if the triage recommendations increase healthcare costs. If GPT4 sends too many patients to urgent care, it may increase costs instead of helping. The authors should measure if AI triage is cost-effective. If they do not they must say in the paper that they did not check if the AI makes healthcare more expensive as limitation of the study.
We agree with the reviewer’s suggestion and have decided to include this into our discussion. Please see lines 270-274.
- There is no discussion of legal risks. If GPT4 makes a wrong triage decision and a patient is harmed, who is responsible? The study does not talk about this. The authors should explain the legal problems of using AI in triage. If they do not they must write that the study does not check the legal risks of AI mistakes.
The reviewer mentions an ethical dilemma worthy of discussion. Thank you for your input, we have added this to our paper to strengthen our discussion. Please see lines 293-298 for a discussion of legal risk with the current use of AI in healthcare triaging and potential future changes.
Reviewer 3 Report
Comments and Suggestions for Authors
Line 6: Leffler MD, MPH (replace and with a comma)
Line 22: remove space between non- urgently
Line 23: specialty vs speciality
Line 35: LLM stands for large language models not learning
Line 41: put a comma after answering, 3. to keep formatting the same
Line 112: GPT-4' is it missing an s after the 4?
Line 166: not sure the exact formatting of this journal but APA recommends spelling out numbers less than 10 and using digits for those greater than 10.
Line 221: all other citations included brackets, 11 is missing them.
----------------------------------------------------------
Comments
Line 85 Its not clear, but I am assuming the 143 messages were all the messages received from Jan 23 - Aug 23, correct? If not, how were those 143 selected?
I think something that could have made this study more robust, would have been the insight from a more seasoned ophthalmologist. I can say after reading a few examples, I would have been more likely to agree with the AI than the doctor in training. Maybe consider the doctor in training, 2 veterans and the AI and compare all the responses. That would be a neat study.
Do these doctors-in-training follow a standard triage protocol? Are there guidelines outlined that they are able to deviate from or was this all based on their personal understanding of standard of care?
Was your LLM pretrained in any way? Beyond the prompts that you provided. Was each case presented with Prompt 1 as a new chat. And each time you started with a "blank slate" or did the model have the opportunity to learn from each prompt. i.e. was prompt engineering a confounding variable that the AI was able to improve upon the responses over time?
Was there ever an attempt to resubmit the patient findings and see if the responses changed in any way? Again a prompt engineering and learning/training questions.
Future study would be neat to compare doctor in training, veteran doctors, and various LLMs such as GROK, ChatGPT, etc. and see how they relate.
Better, with training, update each conversation with the actual diagnosis, outcome and provide any images (de-identified) to train the LLM to improve triage.
Then run new patients and see if they become more aligned with the residents clinical decision making.
Author Response
To the reviewer(s),
Thank you very much for your constructive feedback on our manuscript. We greatly appreciate the time and effort you dedicated to reviewing our work. Your insights have been invaluable in guiding us towards enhancing our manuscript on the Comparative Evaluation of ChatGPT-4 and Ophthalmologist-in-Training in the Triage of Patient Messages Sent to the Eye Clinic via the Electronic Medical Record. We look forward to improving our manuscript for potential publication in your esteemed journal. All new changes in the manuscript can be found in red font.
Sincerely,
Alsumait A, Deshmukh S, Wang C, Leffler C
Comments
1. Line 85 Its not clear, but I am assuming the 143 messages were all the messages received from Jan 23 - Aug 23, correct? If not, how were those 143 selected?
We agree with the reviewer’s suggestion that the language needed clarification. Please see lines 95-97 for updates.
2. I think something that could have made this study more robust, would have been the insight from a more seasoned ophthalmologist. I can say after reading a few examples, I would have been more likely to agree with the AI than the doctor in training. Maybe consider the doctor in training, 2 veterans and the AI and compare all the responses. That would be a neat study.
Here GPT4 was not compared with one or more doctors responding in a simulated or test situation. GPT4's response was compared to the actual response delivered by the practitioners in the department of ophthalmology in real time. The doctors-in-training seek out the attending ophthalmologists for guidance when they have questions. One limitation of our study is that it is not possible to have patients call multiple different university ophthalmology departments to look at the variation in responses between departments, because established patients typically only have a relationship with one ophthalmology practice at a time. See lines 90-93 for updates.
3. Do these doctors-in-training follow a standard triage protocol? Are there guidelines outlined that they are able to deviate from or was this all based on their personal understanding of standard of care?
These were the responses actually delivered by the doctors-in-training based on their training, reading, and judgment, as assisted by the attendings for further guidance when the doctors-in-training had questions. See lines 90-93 for updates.
4. Was your LLM pretrained in any way? Beyond the prompts that you provided. Was each case presented with Prompt 1 as a new chat. And each time you started with a "blank slate" or did the model have the opportunity to learn from each prompt. i.e. was prompt engineering a confounding variable that the AI was able to improve upon the responses over time?
Our LLM was not pretrained in any way beyond the prompts we provided. The messages were entered sequentially one after the other without any additional prompts. The reviewer presents an interesting point that the LLM may have had the oppertunity to learn from each subsequent prompt. As such, we performed another anaysis looking at the percent agreement in the first and second half of the study to see if GPT4 got better over time. For the first 70 patients, the percent agreement on specialty clinic referrals was 63%, and was 61% for the latter half of the cohort. Similarly, for triage acuity, the percent agreement was 64% for the first 70 patient messages and 60% in the second half.
5. Was there ever an attempt to resubmit the patient findings and see if the responses changed in any way? Again a prompt engineering and learning/training questions.
No, there was no attempt to resubmit the patient findings to see if the reponses changed.
6. Future study would be neat to compare doctors in training, veteran doctors, and various LLMs such as GROK, ChatGPT, etc. and see how they relate. Better, with training, update each conversation with the actual diagnosis, outcome and provide any images (de-identified) to train the LLM to improve triage.
We agree with the reviewer’s suggestion. See lines 313-317 for updates.
Round 2
Reviewer 2 Report
Comments and Suggestions for Authors
Thank you for your careful review and updates to the manuscript. I appreciate the effort you put into addressing the comments and improving the study. The new analysis, especially the addition of Cohen’s kappa and the discussion of legal risks, makes the paper stronger. One additional point that I realized later is that the study results are not just about GPT-4, but also about the specific prompt you created and used. A different prompt could give better or worse results, we don’t know. This was something I missed in my first review, and I suggest mentioning this as a limitation in the discussion and the result cannot be generalized to GPT4 but specifically for your prompt. This would make it clear that the AI’s performance depends on how it is asked to triage the cases and could be.
Author Response
Dear Reviewer(s),
Thank you for your thoughtful feedback and for highlighting the influence of the specific prompt used in our study on GPT-4's performance. We agree that the design of the prompt is a critical factor that can significantly affect the outcomes of AI interactions. We will add a section in the discussion to address this point, acknowledging that the results are specific to the prompt used and that variations in prompt design could lead to different performance outcomes. This limitation has been clearly stated in lines 300-305. Your suggestion helps in further refining our study. Thank you again for all your input!
